# ImpText: A Benchmark and Tool-Augmented Framework for Implicit Text Reasoning

**Litao Guo** [* 1]  **Jinsong Zhou** [* 1]  **Shuaibo Li** [1]  **Man CHEN** [1]  **Xinli Xu** [1]  **Zixin Zhang** [1]  **Harold Haodong Chen** [1 2]
**Ying-Cong Chen** [† 1 2]

## Abstract

Multimodal Large Language Models (MLLMs) have demonstrated exceptional proficiency in standard text extraction, but they encounter significant challenges when confronting real-world implicit text. Such content typically contains malicious information, intentionally concealed through physical deformation, visual camouflage, or cognitive suggestion. These concealment techniques circumvent content moderation systems and pose severe risks to user safety. To bridge the research gap in text recognition under real-world adversarial scenarios, we define the task of **Implicit Text Reasoning** and introduce **ImpText-Bench**, a meticulously constructed benchmark. Extensive evaluations on this benchmark reveal significant vulnerability in current systems; even advanced proprietary models achieve a maximum Text Match Score of only 35.79%. In response, we propose **ImpText-Reader**, a tool-augmented framework. It employs a three-stage training strategy utilizing capability-boundary data to collaboratively optimize tool selection and semantic reasoning, thereby effectively extracting hidden text. Extensive experiments demonstrate that our approach achieves SOTA performance, significantly enhancing model robustness in adversarial environments. Project page: ImpText

## 1. Introduction

The rapid evolution of Multimodal Large Language Models (MLLMs) has revolutionized the field of Optical Character Recognition (OCR) and text-rich image understand-

ing. Current systems—including advanced MLLMs like GPT-5 (OpenAI., 2025a) and Gemini-3 (Team, 2025a), as well as specialized architectures (Team et al., 2025; Cui et al., 2025b;a; Li et al., 2025) like DeepSeek-OCR (Wei et al., 2025) have achieved exceptional proficiency in extracting visual text. On established benchmarks such as DocVQA (Mathew et al., 2021), ChartQA (Masry et al., 2022), and OCRBench (Liu et al., 2024d), these architectures have attained performance levels approaching human parity. However, despite their immense success in conventional text extraction, their deployment in real-world content security and moderation systems continues to face severe challenges, characterized by the frequent omission of high-risk target text and significant performance degradation.

This performance gap originates from a fundamental divergence in data characteristics. State-of-the-art models are optimized for **normal text**, implicitly assuming that the primary information is legible, visually salient, and located in the main content area of the image. Consequently, these models tend to prioritize the most distinguishable visual features. However, real-world attackers exploit this bias to deliberately engineer **implicit text**. They conceal high-risk content through physical distortion, visual camouflage, or cognitive suggestion. When identifying such content, models often focus on clear, benign decoy regions, while overlooking malicious content that is visually camouflaged or implied through context. In contrast, humans can effectively recognize such implicit expressions by virtue of contextual associations and adjustable observation methods, such as moving, rotating or tilting screens.

Beyond the performance gap itself, a more severe issue is that most existing benchmarks still focus on explicit text extraction. The academic community currently lacks dedicated benchmarks and methodologies for implicit text recognition. This lack of dedicated benchmarks prevents researchers from clearly understanding the true capabilities of models when confronted with such samples, leading to dilemmas in content moderation for real-world scenarios.

To bridge this critical gap, we first define the task of **Implicit Text Reasoning (ITR)**. We define the ITR task as the challenge of discovering and identifying text that has been in-

---

*Equal contribution  †Corresponding author.  [1]Hong Kong University of Science and Technology (Guangzhou)  [2]Hong Kong University of Science and Technology. Correspondence to: Ying-Cong Chen <yingcongchen@ust.hk>.

*Proceedings of the $43^{rd}$ International Conference on Machine Learning*, Seoul, South Korea. PMLR 306, 2026. Copyright 2026 by the author(s).

tentionally concealed or obfuscated within real-world adversarial scenarios. Based on this definition, we collected raw data from adversarial environments and meticulously constructed **ImpText-Bench**, a benchmark comprising 1,630 high-quality image-implicit text pairs. The dataset was established through a rigorous process involving manual annotation, a regeneration pipeline, and detailed quality checks. Furthermore, we define the Image Classification F1 Score and the Text Match Score (TMS) as core metrics to rigorously evaluate model performance.

Leveraging the ImpText-Bench, we conduct extensive experiments across a wide range of closed-source and open-source models. The results demonstrate substantial limitations in existing systems: even the most advanced proprietary MLLMs and specialized OCR systems achieve a maximum Text Match Score of only **35.79%** when facing implicit text. This finding demonstrates that current models remain highly vulnerable in real-world scenarios and that there is significant room for improvement and optimization.

Furthermore, to address the inadequacy of passive recognition paradigms in handling implicit text under real-world scenarios and enhance the robustness of text recognition systems, we propose **ImpText-Reader**. This framework mimics the human recognition paradigm, shifting from passive identification to an active workflow where suspicious samples are first processed by adaptively invoking image enhancement tools, followed by inferring hidden text based on the enhanced images through an optimized recognizer. We adopt a three-stage training paradigm to collaboratively train the MLLM responsible for tool selection and text recognition reasoning. Additionally, we employ a fine-tuning method on capability-boundary data to expand the performance of both modules. Extensive experiments demonstrate that ImpText-Reader achieves SOTA performance, and ablation studies further validate the effectiveness of our proposed paradigm, training framework, and data selection strategy. We summarize our contributions as follows:

1. We define the Implicit Text Reasoning (ITR) task and construct **ImpText-Bench**, the first benchmark dedicated to evaluating the robustness of systems against implicit text.

2. We conduct large-scale experiments on ImpText-Bench, revealing the prevalent failure of current MLLMs in adversarial environments.

3. We propose the **ImpText-Reader** framework that boosts MLLMs' recognition in real-world adversarial scenarios to SOTA performance, validated via extensive experiments.

**Conflict of Interest Disclosure.** The authors declare no financial conflicts of interest related to this work.

## 2. Related Work

**Visual Text Understanding.** The field of visual text understanding has witnessed a paradigm shift from modularized OCR pipelines (Shi et al., 2016; Zhou et al., 2017; Long et al., 2021; Yin et al., 2024; Cui et al., 2025c; Lyu et al., 2024) to unified architectures driven by MLLMs. Benefiting from massive-scale pre-training, general-purpose MLLMs encompassing both proprietary models (Comanici et al., 2025; Team, 2025a; OpenAI., 2025a; xAI, 2025; OpenAI, 2025; Anthropic, 2025a) and open-source models (Team, 2025b; Wang et al., 2025b; Bai et al., 2025b) have demonstrated strong capabilities in parsing text-rich scenarios. Concurrently, specialized lightweight MLLMs (Blecher et al.; Feng et al., 2025; Li et al., 2025; Cui et al., 2025a) have been developed to enhance efficiency in document parsing and visual text recognition. Despite their proficiency on established benchmarks, SOTA models exhibit severe perceptual degradation when confronted with real-world implicit text data. To address this limitation, we define this challenge as the Implicit Text Reasoning task and introduce a corresponding benchmark to bridge this gap.

**Adversarial Robustness in MLLMs.** Extensive research on MLLM robustness has examined the interplay between adversarial attacks and defenses. Attacks are typically categorized into optimization-based methods using pixel-level noise (Qi et al., 2023; Bailey et al., 2024) and generation-based approaches leveraging typography (Gong et al., 2025; Cui et al., 2024). Corresponding defenses include proactive safety alignment (Zong et al., 2024; Mazeika et al., 2024; Bianchi et al.; Chen et al., 2024; Gou et al., 2025) and reactive external guardrails (Inan et al., 2023; Wang et al., 2024). However, a critical distinction exists regarding the research objective. While existing works prioritize *Safety Alignment* to prevent harmful generation from explicit or gradient-based threats, we focus on *Perceptual Robustness*. We introduce natural semantic adversaries to evaluate the model's capacity to perceive and reason about intrinsically obfuscated information, a domain remaining largely unexplored in current literature.

**Related Benchmarks.** Related benchmarks typically encompass OCR capability evaluations and safety alignment assessments. OCR-related benchmarks (Mathew et al., 2021; Masry et al., 2022; Mathew et al., 2022; Wadhawan et al., 2024; Li et al., 2024; Liu et al., 2024b;e;a; Ouyang et al., 2025; Yang et al., 2025; Fu et al., 2025) rigorously evaluate text extraction and associated reasoning tasks under cooperative settings where visual information is explicitly accessible. These benchmarks, however, presuppose legible inputs and thus overlook the perceptual challenges inherent in real-world adversarial scenarios. In parallel, safety-related benchmarks (Zong et al., 2024; Liu et al., 2024c; Zhou et al.; Helff et al., 2024; Zhang et al., 2025; Wang

et al., 2025a; Hu et al., 2025) concentrate on quantifying model adherence to safety policies against diverse malicious threats. Yet, they primarily target final output compliance rather than identifying perceptual failures induced by implicit samples. Our work pioneers the effort to identify and bridge a specific research void between these paradigms.

## 3. The ImpText-Bench

### 3.1. Problem Formulation

**Motivation.** While MLLMs excel at recognizing standard text, they struggle with real-world adversarial scenarios. The absence of dedicated benchmarks for these conditions prevents quantifying performance or guiding optimization for content security. Moreover, implicit text necessitates a dual-stage evaluation distinct from standard OCR paradigms. Unlike general OCR which typically assumes legibility, implicit text recognition requires a detect-then-recognize cognitive process. Accordingly, we employ a binary classification metric to assess hidden text detection and use edit distance to evaluate the accuracy of content recognition.

**Formal Definition.** We formalize the ITR task as learning a mapping function $f_\theta$ that maps an input image $\mathbf{I}$ to a joint output tuple:

$$f_\theta(\mathbf{I}) = (\hat{y}, \hat{T}) \tag{1}$$

where $\hat{y} \in \{0, 1\}$ denotes the label representing the presence of implicit text and $\hat{T}$ represents the recognized text content.

**Evaluation.** To rigorously evaluate model reliability, we adopt a dual-metric protocol covering existence classification and content recognition. First, for existence classification, we prioritize Recall and F1-score to measure the model's sensitivity to hidden threats. Second, to assess the accuracy of content recognition, we introduce the Text Match Score (TMS). We first compute the Normalized Edit Distance (NED) between the predicted text $\hat{T}$ and ground truth $T^*$:

$$\text{NED}(\hat{T}, T^*) = \frac{\text{Levenshtein}(\hat{T}, T^*)}{\max(|\hat{T}|, |T^*|)} \tag{2}$$

To account for the inherent ambiguity in adversarial scenarios, we relax the strict exact-matching constraint. Specifically, we define a recognition instance as successful (TMS = 1) if the NED remains below a pre-defined tolerance threshold $\tau = 0.5$. This threshold is calibrated to accommodate the visual distortions characteristic of implicit text while preserving semantic fidelity.

### 3.2. Data Construction Pipeline

**Diverse Data Collection.** We collect raw samples of implicit text from Chinese mainstream social media platforms. To ensure content richness, our collection covers diverse risk

entities, encompassing various common contact methods such as hidden URLs, phone numbers, text guiding QR code scanning, and instant messaging accounts. These samples originate from multi-domain scenarios. Each data point is organized as a pair $(\mathbf{I}, T)$, consisting of the adversarial image and the corresponding hidden text content. To ensure ground-truth reliability, all raw samples underwent a rigorous manual annotation process.

**Data Regeneration.** To strictly safeguard user privacy while preserving real-world adversarial distributions, we implement a comprehensive regeneration pipeline leveraging the advanced image generation model Seedream 4.0 (ByteDance, 2025c). This pipeline stratifies raw data into two distinct categories:

- *Implicit Samples:* To generate diverse implicit text samples, we employ a visual disentanglement strategy that decomposes original images into foreground implicit text and background layers. These components are organized into two distinct generative repositories: a text style pool extracted from source images to facilitate style transfer and content rewriting, and a distraction background pool derived either directly from original images or regenerated via textual prompts. Final adversarial samples are synthesized by stochastically sampling and blending components from these repositories, thereby ensuring robust distribution coverage.

- *Benign Samples:* To construct benign samples for the existence classification task, we execute a targeted content erasure process. Utilizing manual bounding box annotations, we excise the implicit text regions and subsequently apply context-aware image inpainting to naturally restore the underlying background textures. This procedure yields clean, non-adversarial images that serve as effective negative samples.

**Data Verification and Splitting.**

Following the regeneration pipeline, we implemented a rigorous three-stage verification ensuring dataset quality and safety. First, during the synthesis phase, we deployed *Doubao 1.6 Flash* (ByteDance, 2025a) to perform real-time automated validation, ensuring that all generated images maintained stylistic coherence and semantic consistency. Subsequently, we conducted a comprehensive manual inspection of all synthesized samples to identify and eliminate instances exhibiting low visual fidelity or discrepancies between the image content and text labels. In the final stage, we applied an occlusion mask to sensitive visual elements, such as QR codes, to prevent potential negative impacts during downstream utilization.

Upon completion of quality assurance, the dataset was partitioned based on collection timestamps, where the training

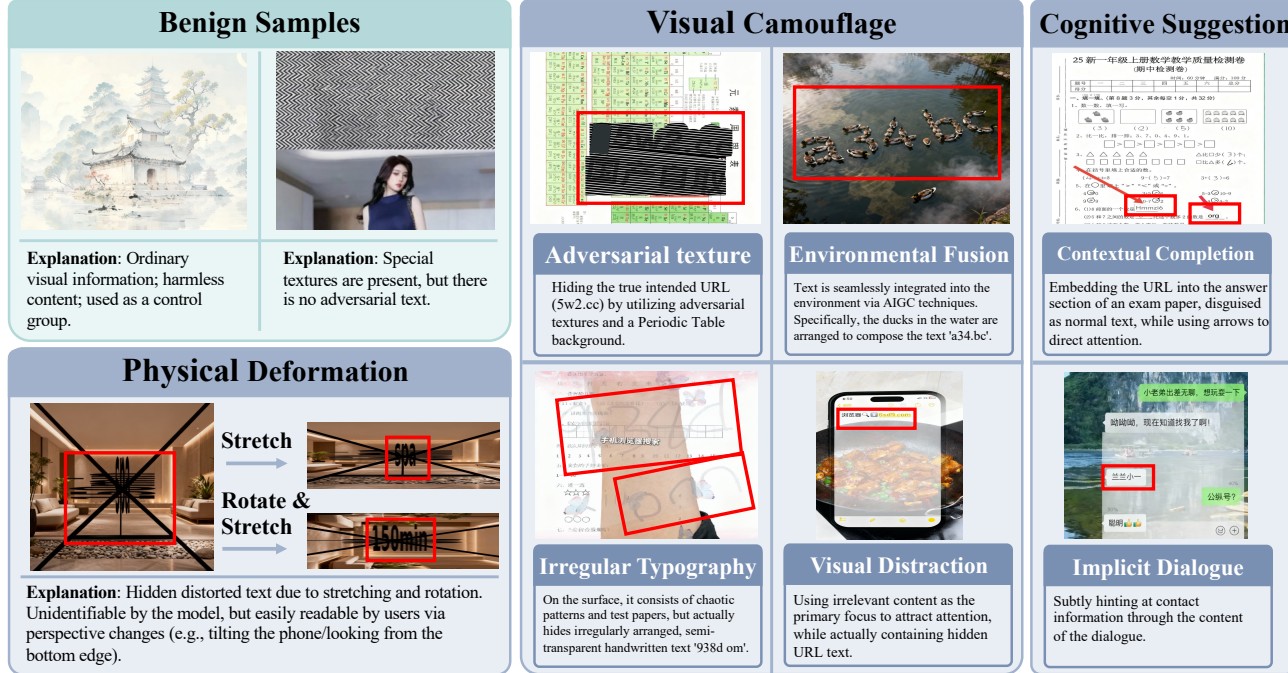

*Figure 1.* **Taxonomy and Representative Examples of ImpText-Bench.** The dataset is organized into four primary categories: (1) **Benign Samples**; (2) **Physical Deformation**, where text is concealed through extreme stretching; (3) **Visual Camouflage**, encompassing sub-types such as Adversarial Texture, Environmental Fusion, and Visual Distraction; and (4) **Cognitive Suggestion**, requiring high-level semantic reasoning to decipher hidden information within contexts like implicit dialogues or contextual completions.

set strictly chronologically precedes the test set. This temporal stratification introduces a natural distribution shift, effectively simulating the real-world evolution of implicit text concealment techniques.

### 3.3. Dataset Statistics and Taxonomy

The ImpText-Bench dataset comprises a total of 1,630 samples. Figure 1 illustrates representative examples from the distinct categories. The statistical distribution and hierarchical structure of the dataset are detailed in Table 1.

**1. Benign Samples.** Real-world content is characterized by class imbalance, where illicit content is sparse within a massive volume of normal data. To simulate this realistic environment, we include 1,141 benign images. These samples contain visible, harmless text or general scenes. This setting poses a challenge for models to minimize false positive rates, requiring the ability to distinguish between genuinely hidden malicious intent and normal visual information.

**2. Physical Deformation.** In such samples, text is typically extremely elongated or compressed along a specific axis. While current OCR and MLLM models often fail to recognize these flattened features, human observers can easily recover the information by physically tilting or rotating the screen to correct the aspect ratio.

**3. Visual Camouflage.** This category encompasses four sub-types. *Environmental Fusion* utilizes generative models to blend text strokes with background elements, making the text appear as a natural part of the scene. *Adversarial Texture* applies complex, noise-like patterns over text regions to disrupt feature extraction. *Irregular Typography* consists of highly distorted handwriting or text constructed from disparate emojis, breaking standard character topology. In *Sticker Occlusion*, implicit text is placed beneath or around prominent, harmless visual subjects.

**4. Cognitive Suggestion.** These samples require high-level semantic reasoning rather than pure pattern recognition. The *Implicit Dialogue* sub-category contains chat logs implying key information, requiring the model to comprehend the conversation flow to detect hidden solicitation. In *Contextual Completion*, contact information is inserted into seemingly benign rich-text images (e.g., test papers).

## 4. The ImpText-Reader

### 4.1. Overview

Current MLLMs encounter significant difficulties when processing implicit text samples in ImpText-Bench. To address this limitation, we draw inspiration from human cognitive

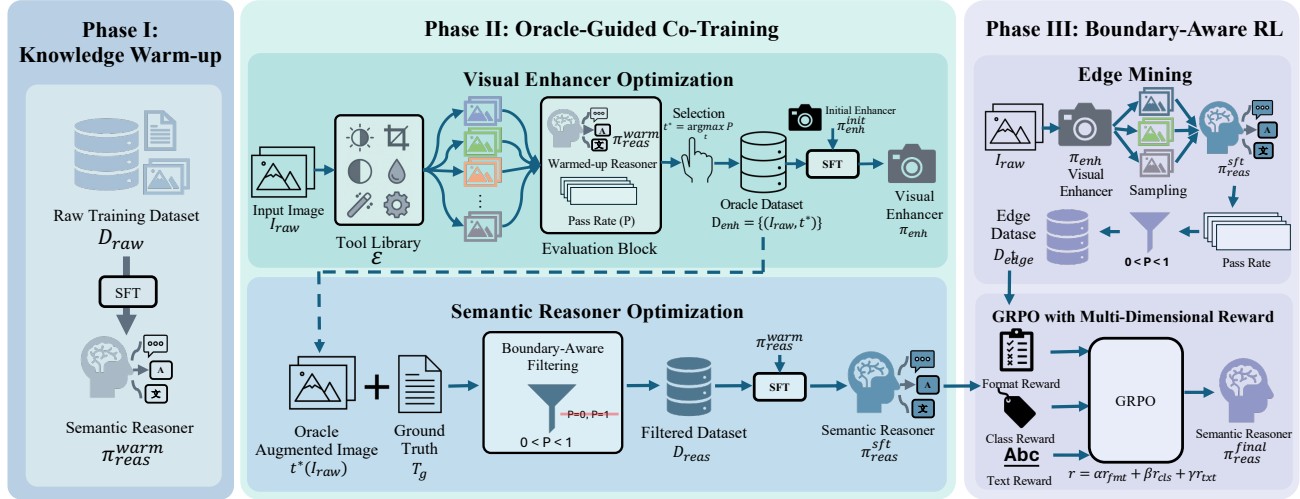

*Figure 2.* **Overview of the ImpText-Reader Training Framework.** The pipeline comprises three progressive stages: (Left) **Phase I: Knowledge Warm-up** fine-tunes the Semantic Reasoner on raw data to establish baseline recognition capabilities. (Middle) **Phase II: Oracle-Guided Co-Training** collaboratively optimizes the Visual Enhancer using oracle tools and the Reasoner using boundary-filtered augmented samples ($0 < P < 1$). (Right) **Phase III: Boundary-Aware RL** robustifies the Reasoner against inference deviations via Edge Mining and multi-dimensional GRPO.

*Table 1.* Detailed statistics of the ImpText-Bench dataset across different concealment taxonomies.

| PRIMARY CATEGORY | SUB-CATEGORY | COUNT |
|---|---|---|
| BENIGN SAMPLES | NORMAL IMAGES | 1,141 |
| PHYSICAL DEFORMATION | STRETCHING | 58 |
| VISUAL CAMOUFLAGE | ADVERSARIAL TEXTURE | 134 |
| | VISUAL DISTRACTION | 93 |
| | ENVIRONMENTAL FUSION | 73 |
| | IRREGULAR TYPOGRAPHY | 28 |
| COGNITIVE SUGGESTION | CONTEXTUAL COMPLETION | 61 |
| | IMPLICIT DIALOGUE | 42 |
| TOTAL | ALL SAMPLES | 1,630 |

patterns. When confronting obfuscated text characterized by adversarial textures or environmental integration, humans subconsciously employ visual enhancement strategies, such as adjusting the observation distance or tilting the display, to disentangle semantic content from visual noise.

Guided by this insight, we introduce ImpText-Reader. This architecture decomposes the Implicit Text Reasoning task into two specialized modules. The **Visual Enhancer** is responsible for adaptive image restoration via tool invocation, while the **Semantic Reasoner** focuses on existence classification and text recognition. However, naive implementation faces distinct hurdles as direct prompting often yields suboptimal tool selection policies, and zero-shot reasoners lack the domain-specific knowledge required to process enhanced imagery. Consequently, we design a progressive three-stage training framework, as illustrated in Figure 2.

### 4.2. Phase I: Knowledge Warm-up

Initially, the Reasoner lacks the fundamental cognitive capability to discern implicit text patterns. To address this, we execute a Cold Start by fine-tuning Semantic Reasoner on the raw, unaugmented dataset $\mathcal{D}_{raw} = \{(I_{raw}, T_{gt})\}$.

$$\pi_{reas}^{warm} \leftarrow \text{SFT}(\pi_{reas}^{init}, \mathcal{D}_{raw}) \qquad (3)$$

This warm-up phase establishes a critical baseline of discriminative capability, enabling the Reasoner to serve as a reliable evaluator in the subsequent co-training phase and preventing training collapse due to low-quality feedback.

### 4.3. Phase II: Oracle-Guided Co-Training

In this phase, we aim to collaboratively optimize the Visual Enhancer to select optimal tools and the Semantic Reasoner to master high-quality inputs.

**Oracle Tool Exploration.** We construct a comprehensive library $\mathcal{E}$ comprising over a dozen image processing tools (see Appendix for details). These tools are designed to augment raw images, rendering indiscernible implicit text explicitly visible. During the data generation stage, we iterate through all tools in $\mathcal{E}$ for each raw image $I_{raw}$ to generate a set of candidates.

These candidates are evaluated by the warmed-up Reasoner $\pi_{reas}^{warm}$ through $k$ inference trials to calculate the recognition *Pass Rate* ($P$). The tool $t^*$ yielding the image with the highest pass rate is identified as the *Oracle Tool*. We construct an instruction dataset mapping raw images to their

optimal tools to fine-tune the Visual Enhancer, enabling it to predict the most effective tool for varying inputs:

$$\mathcal{D}_{enh} = \{(I_{raw}, t^*) \mid t^* = \arg\max_{t \in \mathcal{E}} P(\pi_{reas}^{warm}(t(I_{raw})))\} \tag{4}$$

$$\pi_{enh} \leftarrow \text{SFT}(\pi_{enh}^{init}, \mathcal{D}_{enh}) \tag{5}$$

**Boundary-Aware Data Filtering.** However, for Reasoner, indiscriminate training on all oracle samples is suboptimal. We identify that two categories of data adversely affect model performance: (1) *Trivial samples* ($P = 1$), which represent mastered knowledge with minimal information gain; and (2) *Impossible samples* ($P = 0$), which typically lead to rote memorization rather than logical generalization.

Therefore, we introduce a hardness filter based on the pass rate. We construct a curated dataset $\mathcal{D}_{reas}$ by retaining only samples located within the model's learnable boundary:

$$\mathcal{D}_{reas} = \{(t^*(I_{raw}), T_{gt}) \mid 0 < P(t^*(I_{raw})) < 1\} \tag{6}$$

Using this filtered dataset, we perform SFT on the Reasoner to enhance its semantic recognition capabilities for challenging but solvable samples:

$$\pi_{reas}^{sft} \leftarrow \text{SFT}(\pi_{reas}^{warm}, \mathcal{D}_{reas}) \tag{7}$$

### 4.4. Phase III: Boundary-Aware RL

Phase II assumes ideal tool execution based on oracle knowledge. However, during inference, the Enhancer's prediction may deviate from the optimum. To bridge this gap between training supervision and inference reality, we further optimize the Reasoner using Reinforcement Learning on edge cases generated by the Enhancer's actual policy.

**Edge Mining.** We transition from brute-force search to policy-based sampling. For each raw image, we sample $k'$ distinct tool strategies from the trained Enhancer $\pi_{enh}$. We generate a dataset of Edge Cases by retaining only the augmented instances that fall within the learnable capability boundary:

$$\mathcal{D}_{edge} = \{(t(I_{raw}), T_{gt}) \mid \\ t \sim \pi_{enh}(\cdot|I_{raw}), 0 < P(t(I_{raw})) < 1\} \tag{8}$$

This process explicitly mines scenarios where the Enhancer's output is imperfect, forcing the Reasoner to adapt to realistic noise distributions.

**GRPO with Multi-Dimensional Reward.** We employ Group Relative Policy Optimization (GRPO) to robustify the Reasoner. To provide comprehensive guidance, we first design a composite reward function $r = \lambda_1 r_{fmt} +$

$\lambda_2 r_{cls} + \lambda_3 r_{txt}$. Specifically, $r_{fmt}$ ensures structural adherence to predefined output formats; $r_{cls}$ is a binary reward for accurate existence classification; and $r_{txt}$ offers dense, fine-grained feedback via the continuous metric $1 - \text{NED}(\hat{T}, T_{gt})$.

During training, for each input image $I$, the model samples a group of $G$ outputs $\{o_1, o_2, \ldots, o_G\}$. The advantage $\hat{A}_i$ for each output is computed via group-wise reward normalization:

$$\hat{A}_i = \frac{r_i - \text{mean}(\{r_1, \ldots, r_G\})}{\text{std}(\{r_1, \ldots, r_G\}) + \epsilon} \tag{9}$$

where $r_i$ denotes the composite reward for the $i$-th output. Finally, the policy is updated by minimizing the GRPO loss, which integrates a clipped surrogate objective with a KL-divergence penalty to ensure optimization stability:

$$\mathcal{L}_{\text{GRPO}}(\theta) = -\frac{1}{G} \sum_{i=1}^{G} \left[ \min\left(\rho_i \hat{A}_i, \text{clip}(\rho_i, 1-\epsilon, 1+\epsilon)\hat{A}_i\right) \\ - \beta \text{KL}(\pi_\theta || \pi_{\text{old}}) \right] \tag{10}$$

where $\rho_i = \frac{\pi_\theta(o_i|I)}{\pi_{\text{old}}(o_i|I)}$ represents the probability ratio, and $\beta$ controls the regularization strength to prevent the policy from deviating excessively from the reference model.

## 5. Experiments

### 5.1. Experimental Setup

**Baselines.** To comprehensively evaluate ImpText-Reader, we benchmark against a diverse set of methods spanning four paradigms: (1) **Specialized OCR Systems** (Cui et al., 2025a; Wei et al., 2025) representing the SOTA in traditional text recognition; (2) **Proprietary MLLMs** (Team, 2025a; OpenAI., 2025a;b;c; xAI, 2025; OpenAI, 2025; Anthropic, 2025a;b; ByteDance, 2025a;b), including leading families like GPT, Gemini, and Claude, which establish the current upper bound of general-purpose multimodal capabilities; (3) **Open-Source MLLMs** (Bai et al., 2025b; Team, 2025b; Wang et al., 2025b; Bai et al., 2025a; Wu et al., 2024), exemplified by the Qwen3 and GLM-4 series, representing robust open-source multimodal foundations; and (4) **Fine-tuned Control Groups**, specifically Qwen3-VL 8B variants fine-tuned via standard SFT and GRPO (Shao et al., 2024) on training datasets.

**Metrics.** We adhere to the evaluation protocol established in Section 3. For Classification Quality, we report **Recall** for implicit samples, **Accuracy** for benign ones, and the **F1 Score** to assess the overall safety-usability trade-off. For Text Recognition Quality, we use the **Text Match Score (TMS)** to evaluate the semantic precision of the recovered text.

*Table 2.* **Main Results Comparison on ImpText-Bench.** We evaluate performance across OCR Systems, Proprietary MLLMs, Open-Source MLLMs and Fine-tuned Models. Metrics include Recall and Text Match Score (TMS) for the three adversarial categories, and Accuracy for the Benign samples group. Overall performance is aggregated via F1 Score and TMS. All values are reported in percentage.

| Method | Physical Deformation | | Visual Camouflage | | Cognitive Suggestion | | Benign | Overall | |
| --- | --- | --- | --- | --- | --- | --- | --- | --- | --- |
| | Recall | TMS | Recall | TMS | Recall | TMS | Accuracy | F1 | TMS |
| *OCR Models* | | | | | | | | | |
| PaddleOCR-VL | - | 0.00 | - | 11.28 | - | 0.00 | - | - | 7.57 |
| DeepSeek-OCR | - | 0.00 | - | 1.52 | - | 0.00 | - | - | 1.02 |
| *Proprietary MLLMs* | | | | | | | | | |
| Gemini 2.5 Flash | 10.34 | 0.00 | 28.05 | 15.24 | 21.36 | 12.62 | 93.43 | 35.09 | 12.88 |
| Gemini 3 Flash Preview | 20.69 | 5.17 | 48.48 | 32.62 | 28.16 | 17.48 | 87.64 | 48.19 | 26.18 |
| GPT-5 | 46.55 | 10.34 | 75.30 | 38.72 | 70.87 | 40.78 | 79.23 | 64.68 | 35.79 |
| GPT-5.1 | 37.93 | 8.62 | 61.28 | 33.84 | 71.84 | 33.98 | 84.40 | 61.62 | 30.88 |
| GPT-5.2 | 32.76 | 0.00 | 61.89 | 32.32 | 86.41 | 49.51 | 83.44 | 62.89 | 32.11 |
| Grok-4 Fast | 17.24 | 1.72 | 31.71 | 13.41 | 10.68 | 2.91 | 92.55 | 35.77 | 9.82 |
| OpenAI o3 | 44.83 | 8.62 | 70.12 | 32.32 | 71.84 | 30.10 | 75.90 | 60.33 | 29.04 |
| Claude 3.7 Sonnet | 13.79 | 0.00 | 58.54 | 16.16 | 74.76 | 24.27 | 81.51 | 56.70 | 15.95 |
| Claude 4.5 Sonnet | 18.97 | 0.00 | 57.62 | 14.33 | 75.73 | 23.30 | 92.55 | 65.26 | 14.52 |
| Doubao 1.6 Flash | 10.84 | 2.41 | 44.55 | 31.02 | 50.49 | 33.98 | 94.83 | 52.69 | 26.79 |
| Doubao 1.8 | 69.88 | 3.61 | 72.28 | 27.06 | 76.70 | 30.10 | 74.06 | 62.40 | 23.72 |
| *Open-Source MLLMs* | | | | | | | | | |
| Qwen3-VL 8B | 4.82 | 1.20 | 16.50 | 12.21 | 20.39 | 11.65 | 99.04 | 26.09 | 10.22 |
| Qwen3-VL 30B A3B | 12.07 | 0.00 | 38.72 | 24.70 | 38.83 | 32.04 | 94.30 | 47.80 | 23.31 |
| QVQ 72B Preview | 3.45 | 0.00 | 15.55 | 6.71 | 9.71 | 6.80 | 94.22 | 20.39 | 5.93 |
| GLM 4.6V | 15.52 | 0.00 | 30.79 | 15.85 | 39.81 | 20.39 | 93.60 | 42.36 | 14.93 |
| GLM 4.5V | 3.45 | 0.00 | 14.33 | 11.28 | 23.30 | 20.39 | 98.07 | 25.00 | 11.86 |
| GLM 4.1V 9B | 0.00 | 0.00 | 13.41 | 8.84 | 9.71 | 4.85 | **99.74** | 19.78 | 6.95 |
| Intern-S1 | 20.69 | 6.90 | 33.23 | 18.29 | 41.75 | 26.21 | 94.57 | 45.87 | 18.61 |
| InternVL-3.5 | 39.66 | 1.72 | 57.01 | 28.05 | 56.31 | 36.89 | 91.76 | 62.98 | 26.79 |
| DeepSeek VL2 | 34.48 | 0.00 | 61.28 | 21.34 | 55.34 | 21.36 | 78.53 | 54.94 | 18.81 |
| *Fine-tuned Models* | | | | | | | | | |
| Qwen3-VL 8B + SFT | 71.08 | 15.66 | 64.03 | 43.23 | 53.40 | 38.83 | 96.84 | 73.95 | 37.63 |
| Qwen3-VL 8B + GRPO | 27.71 | 10.84 | 43.23 | 30.69 | 47.57 | 31.07 | 97.72 | 56.55 | 27.40 |
| Ours | **83.13** | **25.30** | **85.15** | **49.83** | **83.50** | **62.14** | 96.76 | **87.97** | **48.26** |

## 5.2. Main Results

As presented in Table 2, our comprehensive evaluation on the ImpText benchmark yields the following key insights:

**Significant Limitations of OCR Models.** OCR models exhibit a marked performance degradation. Constrained by a lack of semantic reasoning capabilities, these models are highly susceptible to interference from complex visual backgrounds. This susceptibility results in predictions that diverge substantially from the ground truth. These findings underscore that explicit text localization and recognition alone are insufficient for implicit text reasoning; rather, a profound semantic comprehension of image content is a prerequisite for deciphering underlying intent and concealed text.

**Performance Bottlenecks in General MLLMs.** While proprietary models excel in general reasoning, Table 2 re-

veals critical perception gaps in adversarial scenarios. Most notably, models are nearly incapable of handling *physical deformations*, with the highest proprietary TMS reaching only 10.34% on extremely stretched or rotated text. Significant fragility persists in *visual camouflage*, where models fail to distinguish text blended into adversarial textures or natural environments. Furthermore, in *cognitive suggestion*, they are easily misled by dominant visual distractors, overlooking subtle hidden text. Consequently, even the SOTA GPT-5 achieves a modest overall TMS of 35.79%. This consistent failure across both proprietary and open-source models confirms that robust implicit text recognition remains a non-trivial, unresolved challenge.

**Effectiveness and Robustness of ImpText-Reader.** Our method achieves SOTA performance across all three implicit text sub-categories. And ImpText-Reader achieves an overall TMS of **48.26%**, surpassing the best fine-tuned

*Table 3*. **Component Analysis of ImpText-Reader.** We report the impact of incrementally adding each module to the baseline model. The results demonstrate that the integration of components progressively unlocks improvements in both F1 Score and TMS.

| Model Configuration | Overall Performance | |
|---|---|---|
| | F1 Score | TMS |
| Qwen3-VL 8B | 26.09 | 10.22 |
| + Tool Augmentation | 28.62 | 10.63 |
| + Knowledge Warm-up | 57.62 | 35.79 |
| + Oracle Tool Exploration | 78.49 | 46.83 |
| **+ Boundary-Aware RL (Ours)** | **87.97** | **48.26** |

*Table 4*. Threshold sensitivity of TMS under different NED thresholds. All values are reported in percentage. Although absolute scores increase as $\tau$ becomes more permissive, the relative ranking remains stable.

| Model | $\tau = 0.1$ | $\tau = 0.3$ | $\tau = 0.5$ | $\tau = 0.7$ | $\tau = 0.9$ |
|---|---|---|---|---|---|
| ImpText-Reader | **22.29** | **34.76** | **48.26** | **59.51** | **72.39** |
| GPT-5 | 14.31 | 25.77 | 35.79 | 47.03 | 58.49 |
| Gemini-3-Flash-Preview | 14.72 | 21.06 | 26.18 | 30.47 | 35.17 |
| GLM-4.6V | 6.54 | 11.45 | 14.93 | 17.59 | 21.47 |
| Qwen3-VL 8B | 6.75 | 8.59 | 10.22 | 11.66 | 13.29 |

baseline and the leading proprietary model GPT-5 by a substantial margin of over **10%**. This significant performance gap highlights the superiority of our approach in handling complex implicit text reasoning. Notably, while substantially improving the recognition accuracy of concealed text, our model maintains a high Benign Accuracy of 96.76%. This indicates that our framework avoids becoming overly sensitive during training, thereby preventing frequent misclassification of benign images. Such balanced performance attests to the robustness of ImpText-Reader and highlights its potential for large-scale industrial deployment.

### 5.3. Threshold Sensitivity Analysis.

Since TMS is computed by thresholding NED, we examine whether our conclusions are sensitive to the choice of $\tau$. We conduct a two-part analysis. First, we evaluate representative models under different thresholds. As shown in Table 4, although the absolute TMS monotonically increases as $\tau$ becomes more permissive, the relative ranking of the models remains stable. In particular, ImpText-Reader consistently outperforms all compared baselines across all threshold values. This demonstrates that our main comparative conclusions are robust and not fragile to the specific choice of $\tau$.

Second, we validate whether $\tau = 0.5$ is a suitable semantic operating point. To this end, we construct a 1,000-sample subset, then use GPT-5 as an impartial judge to determine whether each predicted text preserves the core semantic intent of the ground truth. Finally, we compare this semantic-level judgment with the binary decision induced by thresholding NED at different $\tau$ values. As shown in Table 5,

*Table 5*. Agreement between threshold-based TMS decisions and semantic-level judgments under different NED thresholds. The agreement peaks at $\tau = 0.5$.

| Threshold | $\tau = 0.1$ | $\tau = 0.3$ | $\tau = 0.5$ | $\tau = 0.7$ | $\tau = 0.9$ |
|---|---|---|---|---|---|
| Decision Agreement | 0.840 | 0.886 | **0.909** | 0.899 | 0.827 |

*Table 6*. **Data Strategy Analysis.** Comparison between training on the full dataset versus our boundary-aware strategy. Boundary SFT yields a significant boost in **TMS (+7.77%)**, confirming that filtering out noisy and impossible samples substantially reduces hallucination and improves precise text recognition.

| Sampling Strategy | Overall Performance | |
|---|---|---|
| | F1 Score | TMS |
| Global SFT (Full Dataset) | 76.67 | 39.06 |
| **Boundary SFT (Filtered Data)** | **78.49** | **46.83** |

$\tau = 0.5$ achieves the highest agreement with semantic-level judgments. Overly strict thresholds unfairly penalize minor deviations caused by visual distortions typical in implicit text, while overly loose thresholds incorrectly reward fragmented visible text that fails to convey the hidden message. Therefore, $\tau = 0.5$ is an empirically supported operating point for Implicit Text Reasoning.

### 5.4. Ablation Studies and Analysis

To validate the mechanisms underpinning ImpText-Reader, we conduct comprehensive ablation studies covering component contributions, data sampling strategies, and training paradigms.

**Component Ablation.** Table 3 validates that our performance gains stem from specific training strategies rather than mere tool availability. Initially, simply equipping the baseline with tools yields a negligible improvement (F1: $26.09\% \rightarrow 28.62\%$), underscoring that tools are merely passive instruments; the core lies in the model's ability to wield them. However, as we incrementally integrate Knowledge Warm-up, Oracle Tool Exploration, and Boundary-Aware RL, the performance steadily climbs to a peak of **87.97%**. This step-wise trajectory demonstrates that our framework effectively unlocks the potential of tool usage, progressively expanding the model's capability boundary from passive recognition to active reasoning.

**Data Strategy Analysis.** In Phase II Oracle-Guided Co-Training, we maintain the Visual Enhancer's training data constant while altering the data collection strategy for the Semantic Reasoner. In contrast to our approach, Global SFT trains the Semantic Reasoner using optimal tools across *all* image-tool pairs. Table 6 demonstrates that "Boundary SFT" outperforms "Global SFT" on both metrics, achieving a particularly significant gain of **+7.77%** on the more challenging TMS). This suggests that filtering out noisy or

*Table 7.* **Paradigm Analysis.** Comparison between joint and decoupled optimization. Joint Optimization leads to training collapse due to task interference. In contrast, our Decoupled Optimization isolates perception from reasoning, ensuring superior performance.

| Optimization Paradigm | Overall Performance | |
|---|---|---|
| | F1 Score | TMS |
| Joint Optimization | 21.39 | 8.38 |
| **Decoupled Optimization** | **57.62** | **35.79** |

impossible samples prevents the model from speculative guessing and encourages judgments grounded in precise semantics. This strongly validates our hypothesis that concentrating on boundary training data effectively expands the model's capability frontier.

**Training Paradigm Analysis.** Finally, we investigate the impact of different training paradigms. We compared our approach against "Joint Optimization," where the Visual Enhancer and Semantic Reasoner are treated as a single model engaged in a multi-turn SFT training during the Warm-up phase. Table 7 shows that Joint Optimization leads to severe interference between the tool selection and the text reasoning, resulting in optimization collapse and training failure. In contrast, our Decoupled Optimization strategy successfully separates these tasks, ensuring the convergence of semantic recognition capabilities first, thereby guaranteeing stability in subsequent stages.

## 6. Limitations

While ImpText-Bench establishes a novel foundational benchmark for Implicit Text Reasoning, we acknowledge that the current dataset is predominantly sourced from Chinese social media platforms. The inherent Chinese linguistic context may pose challenges for certain models when processing the samples. Future iterations will expand the benchmark's scope by incorporating multi-lingual implicit text samples, thereby fostering a more comprehensive evaluation.

## 7. Conclusion

This work highlights a critical deficiency in MLLMs: the inability to recognize real-world implicit text. To address this, we define the **Implicit Text Reasoning** task and construct **ImpText-Bench**, a benchmark encompassing physical, visual, and cognitive obfuscations. Evaluations reveal that even advanced models remain fragile in such noncooperative settings. In response, we propose **ImpText-Reader**, a framework shifting from passive extraction to active, tool-augmented reasoning with boundary-aware learning. Experiments demonstrate our method's effectiveness in adversarial scenarios, paving the way for the development

of safer and more robust MLLMs.

## Impact Statement

This work contributes to AI safety by addressing the vulnerability of MLLMs to implicit text. Regarding data ethics, we strictly prioritized user privacy by employing a comprehensive regeneration pipeline. The released benchmark consists solely of synthesized data, ensuring that no personally identifiable information is exposed.

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

# A. Details on Data Synthesis

To strictly safeguard user privacy while preserving the unique stylistic distributions characteristic of real-world implicit content, we implemented a comprehensive regeneration pipeline based on Seedream 4.0 (ByteDance, 2025c). This engine categorizes input samples based on their visual characteristics and dispatches them to specific processing modules. Below, we detail the technical implementation for each core category.

## A.1. Visual Disentanglement and Recomposition

This strategy is primarily applied to **Visual Distraction**, **Implicit Dialogue**, and **Irregular Typography** samples.

The process begins with *Foreground Extraction and Semantic Rewriting*. We utilize the Seedream editor to isolate target objects from the original image and transfer them onto a pure white background. To filter potential risk content and expand dataset diversity, we incorporate a semantic rewriting mechanism: MLLMs first regenerate text that is stylistically consistent but semantically new, followed by Seedream synthesizing the corresponding visual foreground. Since generation models can sometimes fail to render hidden text accurately, we introduce Doubao 1.6 Flash as a discriminator. It scores the stylistic consistency and text legibility between the extracted and original foregrounds, retaining only high-quality samples that exceed a specific threshold.

This is followed by *Background Regeneration and Adaptive Merging*. Simultaneously with foreground extraction, we perform background regeneration on the original image, removing the foreground and restoring occluded textures to build a database of clean backgrounds. Finally, we randomly sample from the foreground and background databases respectively and fuse them using Seedream 4.0. The fused images undergo a final quality assessment by MLLMs to ensure unity in foreground style, text content, and background context, filtering out low-quality artifacts.

## A.2. Geometric Deformation Synthesis

To simulate the **Physical Deformation** category, we implemented a pipeline based on anamorphic illusion logic.

The process involves rendering the target text as a high-resolution bitmap and subjecting it to extreme non-uniform scaling. To construct a difficult-to-discern texture, we generate two layers: Layer A remains at its original angle, while Layer B is rotated. These layers are composited using a multiply blending mode, effectively obscuring the text content when viewed from a normal angle. We then use MLLMs to generate a detailed description of the original background and synthesize a completely new scene via Seedream. Finally, Seedream is used to embed and fuse the deformed foreground text into the generated scene.

## A.3. Environmental Fusion

For the Environmental Fusion category, we employ a fully generative pipeline designed to embed text as an intrinsic part of the scene structure. This process retains no original pixels, preserving only the abstract scene concept.

Doubao 1.6 Flash first analyzes the input image to identify scene semantics or generates a new scene via divergent thinking, subsequently conceiving a specific strategy for concealment. Seedream then generates a brand-new image based on these strategies, ensuring the text is visually integrated into the environment. The final generated images undergo automated quality assessment by MLLMs to filter out failures or samples where the text is too explicit.

## A.4. Adversarial Texture

For text containing complex adversarial texture that are difficult to generate directly, we adopt a strategy that retains the foreground while redrawing the background. First, we use masking to extract the high-frequency textured text regions from the original image. Then, to reconstruct the scene, we use Doubao 1.6 Flash to identify the current background content and generate a new scene description that is stylistically similar but distinct in content. Based on this description, the system generates a completely new background image. Finally, the extracted original textured text is overlaid onto the newly generated background, achieving a seamless integration.

*Table 8.* Comparison with generic tool-use baselines. We report category-level TMS, benign accuracy, token usage, overall F1, and overall TMS. All performance values are reported in percentage.

| Method | Physical Deformation TMS | Visual Camouflage TMS | Cognitive Suggestion TMS | Benign Acc. | Token Usage | Overall F1 | Overall TMS |
|---|---|---|---|---|---|---|---|
| Gemini 3 Flash Preview | 5.17 | 32.62 | 17.48 | 87.64 | 2397 | 48.19 | 26.18 |
| GPT-5 | 10.34 | 38.72 | 40.78 | 79.23 | **474** | 64.68 | 35.79 |
| Gemini 3 Flash Preview + Tool | 5.17 | 12.50 | 7.77 | 95.79 | 5063 | 24.80 | 10.63 |
| GPT-5 + Tool | 12.07 | 31.10 | 29.13 | 93.60 | 2079 | 54.59 | 28.43 |
| ReAct + Tool (GPT-5) | 10.34 | 37.50 | 33.01 | 62.31 | 8006 | 56.63 | 33.33 |
| Best-of-tool (GPT-5) | 5.17 | 38.41 | 34.95 | 87.38 | 6309 | 57.43 | 33.74 |
| Ours | **25.30** | **49.83** | **62.14** | **96.76** | 4570 | **87.97** | **48.26** |

## A.5. Contextual Completion

For Contextual Completion tasks, we employ a local editing strategy. First, we calculate and crop a local context area containing the surrounding environment based on the bounding box. Next, we perform local editing on this region: maintaining the background, font position, size, and style unchanged, we ignore the original text and replace it with new content. Finally, the edited local region is pasted back into the original image to ensure global consistency.

## A.6. Benign Samples

To construct the benign samples, we perform deep erasure on sensitive text regions. The system generates a mask for areas containing text or specific objects and guides the model to remove black occlusions and naturally complete the background, ensuring consistency in color and texture to produce pure samples.

## A.7. Raw Collection and Human Verification

We provide additional details on the construction and filtering process of ImpText-Bench. The raw collection stage was conducted manually between October and November 2025, during which annotators collected candidate images and labeled both the hidden text content and its bounding boxes. In total, we collected 10,626 raw candidate images, most of which contained hidden text such as concealed URLs, phone numbers, and instant-messaging account IDs. All samples were then processed by a privacy-preserving regeneration pipeline, with Doubao 1.6 Flash used for automated validation of image quality, stylistic consistency, and semantic consistency. After removing failed generations and unqualified samples, 4,193 regenerated candidates remained. Three annotators then independently reviewed each candidate, and a sample was retained only if all three unanimously approved its visual similarity, label correctness, and overall image quality. Ultimately, 1,630 samples passed this process and constitute the final ImpText-Bench, corresponding to a 15.3% retention rate from raw collection to final inclusion. We further applied mosaic masking to sensitive visual elements such as QR codes to reduce potential misuse while preserving the intended recognition challenge.

## B. Additional Tool-Use Baselines

To examine whether the gains of ImpText-Reader simply come from access to image enhancement tools, we compare it with generic tool-use baselines using the same tool family. These baselines include strong proprietary MLLMs, a ReAct-style tool-use agent based on GPT-5, and a Best-of-tool baseline. We also report token usage to evaluate whether the improvement is merely caused by a larger inference budget.

The results in Table 8 show that simply adding tools to strong MLLMs does not solve the task and can even hurt performance. For example, Gemini 3 Flash Preview + Tool and GPT-5 + Tool underperform their plain counterparts in overall TMS. Generic agentic baselines also remain below ImpText-Reader: ReAct + Tool reaches 33.33 overall TMS and 56.63 F1, while Best-of-tool reaches 33.74 overall TMS and 57.43 F1. In contrast, ImpText-Reader achieves 48.26 overall TMS and 87.97 F1.

The comparison also shows that the improvement is not explained by token usage. ImpText-Reader consumes fewer tokens than both ReAct + Tool and Best-of-tool, while achieving substantially higher recognition and classification performance. This suggests that the advantage of ImpText-Reader comes from task-specific tool selection and boundary-aware reasoner optimization, rather than from open-ended multi-turn tool use or larger inference budgets.

## C. Qualitative Results

We visualize the comparative results in Figure 3 to intuitively demonstrate the performance of our method against other SOTA MLLMs. The results are color-coded for clarity: **Red** indicates an incorrect prediction regarding the existence of implicit text; **Yellow** represents a correct detection of the Implicit text's presence, but with NED exceeds the threshold of 0.5; and **Green** denotes a successful extraction where the hidden text is correctly detected and NED below 0.5. As demonstrated in the figure, our proposed ImpText-Reader exhibits superior efficacy across various categories, outperforming baseline models in both detection accuracy and text recovery quality.

## D. Hyperparameter Specifications

To ensure reproducibility, we provide a detailed listing of the key hyperparameters used in our framework.

*Table 9.* **Hyperparameters.**

| PARAMETER | VALUE | DESCRIPTION |
|---|---|---|
| *Evaluation* | | |
| NED THRESHOLD ($\tau$) | 0.5 | THRESHOLD FOR DEFINING TMS. |
| *Oracle-Guided Co-Training* | | |
| ORACLE SEARCH TRIALS ($k$) | 8 | NUMBER OF INFERENCE TRIALS PER IMAGE TO IDENTIFY THE OPTIMAL TOOL DURING THE CONSTRUCTION OF $\mathcal{D}_{enh}$ AND $\mathcal{D}_{reas}$. |
| *Boundary-Aware RL* | | |
| EDGE SAMPLING COUNT ($k'$) | 4 | NUMBER OF DISTINCT TOOL STRATEGIES SAMPLED FROM $\pi_{enh}$ TO MINE EDGE CASES FOR $\mathcal{D}_{edge}$. |
| KL COEFF. ($\beta$) | 0.01 | REGULARIZATION WEIGHT FOR THE KL-DIVERGENCE PENALTY. |
| LEARNING RATE | $1e^{-6}$ | LEARNING RATE FOR MODEL DURING GRPO. |

**Rationale for NED Threshold Selection** The selection of the NED threshold $\tau = 0.5$ is grounded in the distinct nature of the Implicit Text Reasoning task. Unlike standard OCR scenarios where input legibility is assumed, implicit text features deliberate visual distortions that inevitably induce character-level noise or punctuation omissions. Stricter thresholds excessively penalize models for minor non-semantic visual artifacts and thereby fail to reflect the true reasoning capability of the model. Conversely, excessively loose thresholds incorrectly incentivize the blind and indiscriminate generation of all visible text. We therefore identified $\tau = 0.5$ as the critical boundary that preserves semantic fidelity. This value ensures that the recognized text conveys the correct malicious intent while simultaneously offering necessary tolerance for local visual perturbations inherent to implicit samples.

## E. Clarifying the Scope of Implicit Text Reasoning

Implicit Text Reasoning (ITR) is related to OCR robustness, but it does not reduce to character-level OCR under stronger visual degradation. Conventional OCR robustness mainly studies whether visible characters can be localized and transcribed under noise, blur, occlusion, distortion, or cluttered backgrounds. In contrast, ITR is defined under an adversarial content-moderation scenario, where high-risk textual information is intentionally concealed, displaced, camouflaged, or implied to evade model-based inspection. Therefore, the goal is not only to transcribe characters, but also to detect whether hidden high-risk text exists and recover the target content from adversarial visual or contextual cues. This distinction is especially clear in cognitive-suggestion samples: all visible text may be legible, yet the challenge lies in identifying which part constitutes concealed contact information, malicious solicitation, or a disguised target string.

This distinction is also supported empirically. On the conventional OCR benchmark OmniDocBench v1.5, PaddleOCR and DeepSeek-OCR achieve high overall scores of 92.56 and 87.36, respectively. However, on ImpText-Bench, their TMS drops to only 7.57 and 1.02, indicating that strong OCR transcription ability is insufficient for this setting. Thus, ITR evaluates a concrete robustness problem that goes beyond conventional OCR transcription.

# F. Visual Enhancer Tool Library Details

This section details the implementation specifications for the 12 image processing tools integrated into the Visual Enhancer module. All tools are implemented using the OpenCV library and are designed to counter specific types of visual obfuscation found in implicit text.

1. **Adaptive Thresholding.** This tool segments text from uneven illumination or complex backgrounds by calculating thresholds locally rather than globally. We convert the image to grayscale and apply the `cv2.adaptiveThreshold` function using the Gaussian method, with an $11 \times 11$ pixel neighborhood block size and a constant subtraction value of 2 to effectively isolate text structures from varying lighting conditions.

2. **Canny Edge Detection.** Designed to extract high-frequency structural outlines while discarding color and texture distractions, this tool is effective for text with low color contrast but distinct boundaries. The implementation involves converting the input to grayscale, applying `cv2.Canny` with a lower threshold of 100 and an upper threshold of 200, and finally inverting the output bits to produce black text on a white background.

3. **Channel Extraction.** This tool is designed to separate text from complex backgrounds by leveraging chromaticity differences rather than luminance. It converts the image to the HSV color space and specifically extracts the S channel. This approach is particularly effective for detecting text that is hidden via color saturation manipulation which might be indistinguishable in grayscale.

4. **JPEG Purify.** To filter out high-frequency adversarial perturbations that are imperceptible to the human eye, this tool utilizes lossy compression artifacts. We simulate a "save and reload" process by encoding the image to a JPEG format memory buffer with a quality factor of 50 (`cv2.IMWRITE_JPEG_QUALITY`) and immediately decoding it back to an image array, effectively discarding high-frequency noise components.

5. **Posterization.** This tool reduces the color space to a few discrete levels, merging gradient backgrounds into solid blocks to highlight text strokes. We apply a Look-Up Table to quantize pixel values $p$ into $N = 4$ discrete levels using the following mapping:

$$p_{new} = \left\lfloor \frac{p}{step} \right\rfloor \times step, \quad \text{where } step = \frac{255}{N}$$

This operation forces subtle color variations in the background to merge while preserving the distinct contrast of the text.

6. **Image Sharpening.** Intended to enhance edge contrast and recover text from blurred inputs like out-of-focus or motion blur, this tool convolves the image with a standard $3 \times 3$ Laplacian-based sharpening kernel $K$:

$$K = \begin{bmatrix} 0 & -1 & 0 \\ -1 & 5 & -1 \\ 0 & -1 & 0 \end{bmatrix}$$

This operation amplifies the differences between the center pixel and its orthogonal neighbors, thereby making character edges more distinct against the background.

7. **Anisotropic Stretch.** This tool counters physical deformation attacks where text is extremely compressed along one axis by generating a composite image containing two aspect-ratio variations. It resizes the image width by $2\times$ for horizontal correction and the height by $2\times$ for vertical correction, then vertically concatenates these two transformed images on a white canvas, allowing the model to scan both potential corrections simultaneously.

8. **CLAHE.** The Contrast Limited Adaptive Histogram Equalization (CLAHE) tool enhances local contrast while preventing noise amplification, making it suitable for images with dark shadows or overexposure. The implementation converts the image to the LAB color space, extracts the lightness channel, applies the CLAHE algorithm with a clip limit of 2.0 and a grid size of $8 \times 8$, and finally merges the processed L channel back with the original A and B channels before converting back to BGR.

9. **Downscale.** Acting as a low-pass filter, this tool removes pixel-level adversarial noise by reducing the image resolution. We resize the image to $0.5\times$ its original width and height using area interpolation (`cv2.INTER_AREA`), which is the method for effectively smoothing out high-frequency noise while retaining macroscopic text structures.

10. **Morphological Closing.** This tool is used to connect broken strokes, such as dot-matrix text, and fill small holes within characters. The process involves inverting the image to treat text as the foreground, applying a morphological closing operation using a $3 \times 3$ rectangular kernel, and then inverting the result back to the original polarity.

11. **Black-hat Transform.** To extract dark text from bright but uneven backgrounds, this tool effectively normalizes lighting variations. We convert the image to grayscale and apply the black-hat operation, defined as the difference between the morphological closing of the image ($I_{raw}$) and the input image itself:

$$I_{blackhat} = \text{Close}(I_{raw}, K_{rect}) - I_{raw}$$

where $K_{rect}$ is a large $15 \times 15$ rectangular kernel. The result is then normalized to the $[0, 255]$ range and inverted to produce dark text on a light background.

12. **Original.** This option preserves the raw input image without any modification. It serves as an identity pass-through operation, selected when the Visual Enhancer determines that the raw image quality is sufficient for recognition or that applying other processing tools might degrade the semantic information by introducing excessive artifacts.

## G. Prompt

The detailed prompt templates and instructions employed in our pipeline.

| Input Image | GT | GPT-5 | Gemini-3-Flash | GLM-4.6V | Ours |
|---|---|---|---|---|---|
| | True
938d com | False
- | False
- | False
- | True
938n com |
| | True
刘览器嗖 ppwr.cc | True
百度，www.cc | False
- | False
- | True
浏览器嗖 ppwr.cc |
| | True
853d con | False
- | False
- | False
- | True
853d com |
| | True
+V 228475 | False
- | False
- | False
- | True
4228475 |
| | True
525 abcd | False
- | False
- | False
- | True
525 abcd |
| | True
困困猪 Zzz E28475 | True
中国联通，12月3日周三·乙巳年十月十四，18:55，我有一个养了二十年的蝴蝶，困困猪 Zzz🦋: E28475，有要看的吗 | False
- | False
- | True
困困猪Zzz E28475
有要 看的吗 |
| | True
938d com | False
- | False
- | False
- | True
9634d com |

*Figure 3.* **Qualitative Comparison Results.**

## Visual Enhancer Prompt

You are an expert in image processing and adversarial text recognition. Your goal is to select the optimal tool from the provided library to reveal hidden text in an image.

The available tool library is defined as follows (please analyze based on this list):
1. adaptive_thresholding: Adaptive Thresholding. Calculates thresholds based on local lighting; suitable for removing complex background textures and extracting text skeletons.
2. canny_edge_extraction: Canny Edge Detection. Discards color and texture, extracts only high-frequency contours, and inverts them to black lines on a white background; suitable for cases with clear text contours but low contrast.
3. channel_extraction: Channel Extraction. Separates R/G/B or Saturation (S) channels; suitable when text exists only in a specific color channel.
4. jpeg_purify: JPEG Re-encoding. Destroys high-frequency micro-perturbations through lossy compression; suitable for adversarial examples or hidden information based on high-frequency noise.
5. posterization: Posterization. Drastically reduces the number of colors (quantization); suitable for merging gradient backgrounds into single color blocks to highlight text.
6. sharpening: Image Sharpening. Enhances image edge contrast; suitable for blurred text.
7. anisotropic_stretch: Anisotropic Stretch. Stretches the image aspect ratio; suitable for identifying extremely flattened or compressed/distorted text.
8. clahe: CLAHE. Enhances local area contrast; suitable for uneven lighting or low-contrast images.
9. downscale: Downscaling (2x). Reduces resolution; suitable for using the blurring effect to filter out high-frequency adversarial noise.
10. morphological_closing: Morphological Closing. Dilation followed by erosion; suitable for repairing dot-matrix patterns or connecting broken text strokes.
11. blackhat_extraction: Black-Hat Transform. Removes light background shading through morphological difference; specifically used to extract dark text on uneven backgrounds.
12. original: Original Image. No processing; suitable when the text is already visible or other tools introduce too much distortion.

# Task
Analyze the visual features (noise, blur, distortion) and select the ONE best tool that is most likely to reveal the hidden text.

# Output Format
Strictly follow this format:
<tool> [Tool Name] </tool>

## Semantic Reasoner Prompt

Analyze the image to determine if it contains any hidden text, which may include contact details, URLs, or other concealed messages.
Your final output must indicate \"Yes\" or \"No\" and provide the content of the hidden text found.
Think step-by-step. Please strictly adhere to the following output format:
<think> [Your reasoning process] </think>
<has_hidden_text> [Yes/No] </has_hidden_text>
<hidden_content> [The extracted text or \"None\"] </hidden_content>

