# OpenReview forum: "ImpText: A Benchmark and Tool-Augmented Framework for Implicit Text Reasoning"
_ICML.cc/2026/Conference — ICML 2026 regular_

### Official Review · Reviewer_8FAS · 2026-03-10

**Soundness:** 3
**Presentation:** 3
**Significance:** 2
**Originality:** 3
**Overall Recommendation:** 5
**Confidence:** 3

**Summary:**

1. This paper focuses on a critical topic of Multimodal Large Language Models in content safety: while current MLLMs excel at standard OCR tasks, they fail significantly when malicious actors intentionally obfuscate illicit information (e.g., scam URLs, contact numbers) using extreme physical distortion, visual camouflage, or cognitive suggestion.
2. This paper proposes a benchmark to evaluate current mainstream MLLMs, identifies significant shortcomings, and then designs a method to train a models to alleviate this issue.

**Compliance With Llm Reviewing Policy:**

Affirmed.

**Key Questions For Authors:**

1. The threshold selection \tau  requires more ablation experiments to verify its sensitivity.

**Limitations:**

Yes.

**Strengths And Weaknesses:**

1. Soundness

Strengths: This paper investigates the issue of implicit text in OCR applications for multimodal large language models and introduces ImpText-Bench. I believe this problem holds significant practical importance. The authors provide a comprehensive account of data construction, quality verification, evaluation experiments, methodological design, and ablation studies, presenting a relatively complete body of work.

Weaknesses: Looking at the overall pipelin, the threshold selection \tau is critical. Although the authors provide their rationale in the appendix. However, if I haven't missed anything, I did not find experiments validating the sensitivity of this threshold.

2. Presentation

Strengths: The overall structure of the paper is quite clear. It first poses the problem, then introduces the benchmark, and finally presents the method and experiments, making for a smooth read.

Weaknesses: The details in Figure 2 are overly complex, making it difficult to grasp the main point at first glance.


3. Significance

The problem addressed in this paper is meaningful, as implicit text is indeed relevant to practical scenarios such as content security and bypassing moderation. The experimental results also demonstrate that existing models still have notable shortcomings on this task.


4. Originality

The paper's main innovation lies in its problem definition and benchmark. This is novel. In comparison, the methodology itself appears more like a combination of existing ideas, such as tool augmentation, SFT, oracle-guided training, and RL, so the originality of the approach is moderate. Overall, this is an innovative paper.

---

> ### Author Rebuttal · Authors · 2026-03-30
>
> We sincerely thank you for the positive assessment of our benchmark's novelty and practical importance, as well as the paper's overall completeness. Below we address your specific feedback.
>
> **Q1: The threshold selection τ requires more ablation experiments to verify its sensitivity.**
>
> **Response:** We agree that the threshold τ requires explicit empirical justification. We have conducted a two-part sensitivity analysis which will be added to the revised manuscript.
>
> **1. Robustness of the Main Conclusions:**
> We evaluated the Text Match Score across representative thresholds:
>
> | Model | τ=0.1 | τ=0.3 | τ=0.5 | τ=0.7 | τ=0.9 |
> | :--- | :--- | :--- | :--- | :--- | :--- |
> | ImpText-Reader | 22.29 | 34.76 | 48.26 | 59.51 | 72.39 |
> | GPT-5 | 14.31 | 25.77 | 35.79 | 47.03 | 58.49 |
> | Gemini-3-Flash-Preview | 14.72 | 21.06 | 26.18 | 30.47 | 35.17 |
> | GLM-4.6V | 6.54 | 11.45 | 14.93 | 17.59 | 21.47 |
> | Qwen3-VL 8B | 6.75 | 8.59 | 10.22 | 11.66 | 13.29 |
>
> While the absolute score monotonically increases as τ loosens, the relative ranking of the models remains stable. This demonstrates that our comparative conclusions are robust and not fragile to the specific choice of τ.
>
> **2. Justification of the Optimal Semantic Operating Point:**
> To validate our statement in the paper that a threshold of 0.5 best captures semantic fidelity, we measured how well the threshold decision aligns with semantic-level judgments. To form a 1,000-sample subset, we randomly selected 250 prediction-ground truth pairs from the outputs of four representative models: ImpText-Reader, GPT-5, Gemini-3-Flash-Preview, and GLM-4.6V. We then used GPT-5 as an impartial judge to determine if the predicted text preserved the core semantic intent of the ground truth. Finally, we compared this semantic judgment against the binary thresholding:
>
> | Threshold | τ=0.1 | τ=0.3 | τ=0.5 | τ=0.7 | τ=0.9 |
> | :--- | :--- | :--- | :--- | :--- | :--- |
> | **Decision Agreement** | 0.840 | 0.886 | **0.909** | 0.899 | 0.827 |
>
> The threshold of 0.5 achieves the highest alignment with semantic-level judgments. Overly strict thresholds unfairly penalize minor visual distortions typical in implicit text, while overly loose thresholds incorrectly reward the generation of fragmented visible text that fails to convey the hidden message. Thus, 0.5 is an empirically supported optimal operating point for the Implicit Text Reasoning task.
>
> **Q2: The details in Figure 2 are overly complex, making it difficult to grasp.**
>
> **Response:** We agree that Figure 2 is visually dense. For the camera-ready version, we will redesign the figure to clearly separate the three training phases and strengthen the visual hierarchy, making the core methodology immediately intuitive at a glance.
>
> **Q3: Methodology originality is moderate, acting more as a combination of existing ideas.**
>
> **Response:** We appreciate this balanced assessment. We agree that the primary contribution of this work lies in the problem definition, the benchmark construction, and revealing the real-world vulnerabilities of current multimodal models. Our method serves more as a strong task baseline while also incorporating key designs tailored to this task, such as boundary-aware filtering and decoupled optimization. These provide a solid starting point for future research in this domain.

---

### Official Review · Reviewer_wkMW · 2026-03-11

**Soundness:** 2
**Presentation:** 3
**Significance:** 2
**Originality:** 2
**Overall Recommendation:** 2
**Confidence:** 4

**Summary:**

This submission provides two contributions: 1). a benchmark dataset for implicit text reasoning; 2). a pipeline of training based (SFT+ RL) to address the implicit text reasoning problem. Extensive results show the effectiveness of the proposed pipeline for the implicit text reasoning job.
Basically, the idea is not new, the proposed new term "implicit text reasoning" actually is still the well-defined task OCR. That is the hard case of OCR. From this point of view, the dataset is of limited novelty.
Moreover, the proposed agentic solution with SFT and agentic RL used well-known techniques, the methodological novelty is also limited.

**Compliance With Llm Reviewing Policy:**

Affirmed.

**Final Justification:**

The authors are confusing the task objective with the method of solving the problem. The objective of this task is robust OCR; resisting adversarial examples is not a new objective. I strongly object to the definition of so-called “new tasks.” The authors refer to a “reasoning aspect,” but this pertains to the method of solving the problem, not whether the objective itself is a new one. I believe we should not encourage this kind of arbitrary redefinition of new problems. Thus, i will keep my negative score.

**Key Questions For Authors:**

1. Are there agentic solution (LLM with reasoning and tool call) to address hard cases of OCR task?

2. What if we use strong existing MLLM to build the agent with took call and reasoning, what would the results be (same token budget usage)?

**Limitations:**

Yes

**Strengths And Weaknesses:**

Pros:
1. The work is well-structured in writing, thus it is easy to follow.
2. The expereimental study is somehow thorough. Basically, it compared different LLM models.

Cons:
1. For the benchmark part, the contribution is limited. Bascially, the paper tried to propose a new task "define the task of Implicit
Text Reasoning". However, it is also the hard cases of traditional OCR. I don't think this is a totally new task.

2. The dataset provided is small. The majority cases are benign cases.

3. The proposed SFT+agentic RL pipeline is also with limitted novelty. The techniques are existing well-known ones.

4. The comparison is not fair in the main results table. The baselines are MLLM models. However, the proposed method is agentic solution. The token usage is quite different. It lacks of comparison to other agentic solutions for OCR with tool use.

---

> ### Author Rebuttal · Authors · 2026-03-30
>
> We sincerely thank you for the feedback. Below we address each point in turn.
>
> **Q1: "Implicit Text Reasoning" is essentially hard OCR**
>
> **Response:**
>
> We believe that implicit text reasoning (ITR) cannot be reduced to merely a harder form of OCR.
>
> From data perspective, difficult OCR samples still contain explicitly present and visually accessible text, although recognition is hindered by blur, occlusion, distortion, or complex backgrounds. In contrast, ITR involves text that is deliberately concealed in realistic adversarial contexts, such as a disguised malicious URL, an extremely stretched string, or text fused with adversarial textures.
>
> This difference in data form directly leads to OCR failure. On the conventional OCR benchmark OmniDocBench v1.5, PaddleOCR and DeepSeek-OCR achieve overall scores of 92.56 and 87.36, respectively. However, on ImpText-Bench, their TMS drops to only 7.57 and 1.02, indicating that they are essentially ineffective in this setting. Such a dramatic collapse is difficult to explain as merely harder OCR. Together, these differences indicate that ITR should be treated as a distinct task rather than hard OCR.
>
> **Q2: The dataset is small, and most samples are benign**
>
> **Response:**
>
> We appreciate this concern. In terms of scale, related benchmarks such as VLSBench (2.2k samples) and OCRBench (1k samples) are comparable to ours. More importantly, ImpText-Bench prioritizes curation over scale. Each sample comes from real adversarial content and is filtered through manual annotation, MLLM quality checks, and final human verification. The 15.3% retention rate from 10,626 collected samples to 1,630 final samples reflects this strict quality control.
>
> The relatively high proportion of benign samples is intentional. Real-world auditing is inherently imbalanced, with benign content far outnumbering non-compliant cases, so such samples are necessary for realistically evaluating false-positive control. Prior work and extensive empirical evidence show that without such benign samples, models tend to rely on spurious correlations in objects or backgrounds rather than learning the actual target, which harms generalization. This design therefore makes the benchmark more reliable for evaluating true recognition and false-positive control.
>
> **Q3: Pipeline has limited methodological novelty**
>
> **Response:**
>
> We agree that our work does not claim a new general-purpose SFT or RL framework. Instead, ImpText-Reader is intended as a strong method that also serves as a strong baseline for future ITR research. Its novelty lies in being specifically designed for the unique characteristics of ITR, with a dedicated task decomposition and a progressive training pipeline. More importantly, our ablation studies show that the gains do not come from simply combining existing techniques. Tool augmentation alone yields only limited improvement, while our Boundary SFT outperforms direct SFT. Therefore, we believe the methodological novelty of ImpText-Reader lies in its task-specific modeling and training design for ITR, rather than in the isolated use of SFT, tools, or RL.
>
> **Q4: Agentic baselines with tools**
>
> **Response:**
>
> To address the fairness concern on tool-based baselines, we add experiments using the same tool family as our method. The table compares strong MLLMs and generic agentic baselines, while also reporting token usage.
>
> |Method|Physical Deformation (TMS)|Visual Camouflage (TMS)|Cognitive Suggestion (TMS)|Benign (Accuracy)|Token Usage|Overall F1|Overall TMS|
> |:---|:---:|:---:|:---:|:---:|:---:|:---:|:---:|
> |Gemini 3 Flash Preview|5.17|32.62|17.48|87.64|2397|48.19|26.18|
> |GPT-5|10.34|38.72|40.78|79.23|**474**|64.68|35.79|
> |Gemini 3 Flash Preview + Tool|5.17|12.50|7.77|95.79|5063|24.80|10.63|
> |GPT-5 + Tool|12.07|31.10|29.13|93.60|2079|54.59|28.43|
> |React + Tool (GPT-5)|10.34|37.50|33.01|62.31|8006|56.63|33.33|
> |Best-of-tool (GPT-5)|5.17|38.41|34.95|87.38|6309|57.43|33.74|
> |Ours|**25.30**|**49.83**|**62.14**|**96.76**|4570|**87.97**|**48.26**|
>
> As shown, simply adding tools to strong MLLMs does not solve the task and may even hurt performance: both Gemini 3 Flash Preview + Tool and GPT-5 + Tool underperform their plain counterparts. Agentic baselines also remain below our method: ReAct+Tool achieves 33.33 TMS / 56.63 F1, and Best-of-Tool achieves 33.74 / 57.43, while ours reaches 48.26 / 87.97. This suggests that tool use may introduce noise or shift images away from the model’s familiar distribution.
>
> The table also addresses token usage. ImpText-Reader is a fixed two-stage pipeline, not an open-ended multi-turn agent. Its token cost is comparable to other baselines and lower than both ReAct + Tool and Best-of-Tool. Our advantage thus comes from a better effectiveness and efficiency trade-off, not higher token use.
>
> Overall, these results show that tool use and agentic prompting are insufficient for this task, while our task-specific design is clearly more effective.

---

> > ### Author Rebuttal · Reviewer_wkMW · 2026-04-01
> >
> > I disagree with the assertion that “implicit text” falls outside the scope of OCR; it should be considered part of OCR robustness. Adversarial samples also fall under this category. As long as the text is visible in the image, it falls within the scope of OCR. Unless it is not visible—in which case it would be a form of steganography—it does not fall within the scope of OCR. The author has coined a new term, “implicit text,” in an attempt to establish a new research topic, but I have reservations about this. This actually falls within the scope of OCR robustness.
> > The concern regarding the small dataset has not been addressed.
> > The issue of the pipeline having limited methodological novelty has also not been resolved.
> > I thus would keep my origianl score.

---

> > > ### Author Response · Authors · 2026-04-02
> > >
> > > We thank the reviewer for their continued discussion and address the three remaining concerns below.
> > >
> > > ## Q1: Whether Implicit Text Reasoning is merely OCR robustness
> > >
> > > The reviewer argues that visible text belongs to OCR and invisible text to steganography. We respect this view but believe **this binary does not account for the "Reasoning" dimension** central to our task definition and paper title.
> > >
> > > ITR does involve text in images and overlaps with OCR, but its core challenge **goes beyond recognition**. As formalized in Section 3.1, ITR requires **reasoning on top of recognition**: determining which text constitutes hidden malicious information and how the adversarial intent is constructed. This is not a difference in degree or difficulty, but **a difference in the nature of the challenge**.
> > >
> > > Consider the Cognitive Suggestion category in Figure 1. Contextual Completion samples embed malicious URLs into exam answer sections disguised as normal text. Implicit Dialogue samples hint at contact information through ordinary conversation. In both cases, **all text is completely legible** — an OCR system can accurately extract every character, yet it has no way to determine which portions constitute hidden malicious content. **No amount of OCR robustness improvement can address a challenge that lies entirely outside the recognition pipeline.**
> > >
> > > This reasoning demand pervades **all ITR categories**. Visual Distraction uses decoys to divert attention, requiring the model to distinguish threats from distractors. Environmental Fusion arranges natural scene elements into text, requiring inference that objects encode information. Physical Deformation requires inferring appropriate perspective transformation strategies. These demands go well beyond "recognizing text under difficult conditions."
> > >
> > > Table 2 provides direct experimental support. PaddleOCR-VL, a specialized OCR system, scores **0.00% TMS on Cognitive Suggestion** — the category where text is most clearly visible — yet achieves **11.28% on Visual Camouflage** where text is genuinely harder to see. This **performance inversion across categories** is inexplicable under an OCR robustness framing, yet directly illustrates the importance of reasoning in the ITR task.
> > >
> > > We also wish to emphasize that whether ITR is labeled "a new task" or "OCR robustness" **does not change the factual contributions of this paper**. Regardless of naming, we are the first to reveal the fragility of SOTA systems on these samples, have carefully constructed a benchmark to enable systematic evaluation, and our method significantly improves performance. Defining dedicated benchmarks for specific challenging scenarios is standard academic practice, and the contributions should be evaluated on their own merits, not through the lens of naming conventions.
> > >
> > > ## Q2: Dataset size
> > >
> > > Our previous response addressed this concern in detail. The reviewer states this remains unresolved but does not identify which aspects were insufficient, nor introduce new arguments.
> > >
> > > Nevertheless, we provide additional context. Real-world adversarial content on social media is **inherently scarce and difficult to collect at scale**. From 10,626 raw images, multi-stage filtering with unanimous agreement among three independent annotators per sample yielded a **15.3% retention rate**, reflecting the rigor of our quality standards rather than a limitation. Benchmarks at this scale are standard and widely recognized in the adversarial and safety domain, including VLSBench (2.2k) and OCRBench (1k). We welcome specific criteria from the reviewer that would help us respond more constructively.
> > >
> > > ## Q3: Methodological novelty
> > >
> > > As stated in our previous response, the primary contribution of this paper is **problem definition and benchmark construction**, with ImpText-Reader positioned as a strong baseline designed for this task. The reviewer's second-round comment on this point again does not introduce new arguments.
> > >
> > > We wish to reiterate that the novelty of our method lies in its **design tailored to the unique characteristics of ITR**, involving multiple carefully motivated choices, each empirically validated. Moreover, our ablation studies reveal insights with practical implications for future ITR research: naively equipping strong MLLMs with tools **does not yield meaningful improvement and can even hurt performance** (as shown in our supplementary experiments), Boundary SFT **significantly outperforms** Global SFT (Table 4), and Joint Optimization **causes training collapse** (Table 5). These findings offer concrete guidance for subsequent work on this task.
> > >
> > > ## Summary
> > >
> > > The reviewer raised five concerns in the first round. The concerns about agentic baselines and token usage were addressed through supplementary experiments in our previous response and were not re-raised. The remaining three are addressed above. We look forward to further discussion.

---

### Official Review · Reviewer_E5tK · 2026-03-13

**Soundness:** 2
**Presentation:** 3
**Significance:** 3
**Originality:** 3
**Overall Recommendation:** 5
**Confidence:** 3

**Summary:**

The paper proposes a new problem statement: Implicit Text Reasoning, which aims to recognize and detect text that is hidden or present in an obscure manner in an image. To support this problem statement, the paper creates a new benchmark, called ImpText-Bench, and shows that existing multimodal LLMs (MLLMs) do quite badly on this benchmark. Furthermore, the paper finally proposes ImpText-Reader, a pipeline to train models that perform the task better than existing MLLMs, and show improvement in performance.

The paper uses raw data collection from actual images on social media, followed by MLLM-supported semi-manual editing of these images to create adversarial samples, and finally MLLM-based and manual filtering, to create the ImpText-Bench dataset. For the ImpText-Reader pipeline, the paper combines using the training data directly for SFT, using the same data to 'choose' between various image-enhancing tools and then using the enhanced version to find text, and finally, using this data to perform an overall GRPO-based fine-tuning. The main takeaways from the paper are empirical results, showing that existing MLLMs don't do well on ImpText-Bench, while the pipeline proposed in the paper is able to achieve far better results. The paper also performs an ablation study to show that various components of their pipeline are indeed helpful in achieving the performance gains.

**Compliance With Llm Reviewing Policy:**

Affirmed.

**Final Justification:**

The authors have addressed most/all of my concerns, and I've increased the score accordingly.

**Key Questions For Authors:**

1. Can you please provide more explicit details on the data generation process, specifically for all steps in which human intervention was used to make sure the dataset is of high quality?
2. Can you please properly describe the 'attack scenario' here, and why Implicit Text Reasoning is important to protect against such an attack?

**Limitations:**

yes

**Strengths And Weaknesses:**

Soundness
-------------

The dataset ImpText-Bench is generated using MLLMs, and thus, additional checks and balances to make sure there are no errors introduced into the dataset are very important. The pipeline to generate the dataset in the paper does include these checks, which is good. However, unfortunately, information about these steps is lacking, which makes it difficult to judge the quality of the dataset. More specifically, the data creation process starts with raw data collection that contains hidden URLs, phone numbers, etc., but there is a lack of information on how exactly this data was collected, how many images were collected at this stage, etc. This dataset is then passed through a 'data regeneration' pipeline to maintain the style while changing the content to maintain the privacy of the original data. Finally,  to make sure the generated data from this pipeline is of high quality, there is a manual filtering done, but again, this step lacks information on how this filtering was done, whether it was just one annotator per image, how many images were filtered out in this step, etc.

It should also be mentioned here that the dataset/benchmark is not uploaded as supplementary material. I assume it will be released with the paper, if the paper is accepted, but for now, the reviewers don't have the ability to inspect the dataset.

The ImpText-Reader aspect of the paper, on the other hand, looks sound to me, and the pipeline seems to indeed improve the performance for Implicit Text Reasoning.

Presentation
--------------
I found the presentation of the work good, and mostly easy to follow. No comments here.

Significance
--------------

Implicit Text Reasoning, if seen from the perspective of robustness in text reasoning in MLLMs, is a valuable task. The dataset generated in this work, as well as the training pipeline, thus has significant value for the community.

However, the framing around 'attackers' confuses me a little. This is important because many of the techniques discussed to create this benchmark involve things that are not natural changes, but something that needs to be done intentionally to hide information (eg, extending the text to unnaturally high ratios). Can the authors provide more information on the exact scenario here? Who exactly is the attacker, whose aim it seems is to fool the MLLM into not detecting the implicit text present in the image? In what scenario would this be problematic/harmful?

Originality
------------

I do not work actively in the field of text reasoning from images, so my judgment of originality is based on the related work discussion provided in the paper. Based on that, it does seem like both the benchmark created in the paper, as well as the pipeline proposed to perform Implicit Text Reasoning, are novel contributions.

Overall, I find the work valuable. I am currently only recommending a Weak Accept, as I would appreciate some clarification of the points raised in my review, but I believe if those concerns are appropriately addressed, this would be a good paper, and I would increase my rating to Accept.

---

> ### Author Rebuttal · Authors · 2026-03-30
>
> We sincerely thank you for the positive evaluation of our benchmark and methodology, and for the valuable time and effort dedicated to reviewing our work. Below, we provide the detailed information requested.
>
> **Q1: Details on the data generation process and human intervention.**
>
> **Response:**
>
> Throughout the data generation process, human intervention primarily occurred in raw data collection and annotation, post-regeneration quality inspection, and final safety filtering.
> * **Raw Collection:** The collection of our raw data was completely manual. Between October and November 2025, we collected candidate images from social media platforms. During this stage, annotators simultaneously labeled the hidden text content and the corresponding bounding boxes within the collected samples. The majority of the candidates collected at this stage already contained hidden text, such as concealed URLs, phone numbers and instant messaging account IDs. Following this stage, we conducted a detailed analysis of the collected samples, established the taxonomy for ImpText-Bench, and determined the subsequent regeneration pipelines for different categories.
> * **Manual Quality Inspection:** After the data regeneration process, three annotators independently reviewed each remaining sample. A sample was retained only if all three annotators unanimously approved its visual similarity to the original image, the correctness of the text label, and the overall image quality. Finally, we applied safety post-processing to the retained images by applying mosaic masking to sensitive visual elements (such as QR codes) to prevent potential misuse.
> * **Data Statistics:** For this benchmark, we collected a total of 10,626 raw images. Most of these images contained hidden text. All samples then went through a privacy-preserving regeneration pipeline. During regeneration, Doubao 1.6 Flash performed automated validation to check image quality, as well as stylistic and semantic consistency with the original images. After automatically filtering out failed generations and unqualified images, 4,193 images remained. At this stage, the images included both samples with hidden text and Benign samples. Ultimately, 1,630 samples passed the unanimous quality review by the 3 annotators during the manual quality inspection, and these images constitute the final ImpText-Bench.
>
> **Q2: Dataset upload and availability.**
>
> **Response:**
> We greatly appreciate your understanding. We commit to fully and publicly releasing the ImpText-Bench benchmark upon acceptance of this paper.
>
> **Q3: Clarification of the attack scenario and the importance of Implicit Text Reasoning.**
>
> **Response:**
>
> We would like to further clarify that the setting discussed in our paper is not a natural degradation or natural noise scenario, but rather an intentional, adversarial content safety evasion scenario.
>
> Specifically, the attacker is malicious content publishers attempting to disseminate textual violating content. These contents typically direct users to illegal websites, provide contact information for illegal services, or point to the accounts of cyber scammers. The defender is an automated content moderation system powered by MLLMs deployed on social media platforms. To evade MLLM-based content moderation, attackers intentionally use various concealment techniques to obscure key information. Once these hidden contents successfully bypass the moderation system, they can directly reach a massive base of end-users. Directing users to cyber scams or illegal services could lead to severe financial losses and privacy breaches for end-users, while also undermining the trust of the online community and the health of the platform's ecosystem.
>
> More importantly, the raw samples we used for analysis were all manually collected from real social media platforms. This demonstrates that the aforementioned concealment strategies are not merely theoretical assumptions, but rather attack methods already widely utilized in real-world scenarios. Therefore, there is a strong, practical necessity to enhance the defense and recognition capabilities of models under such realistic attack scenarios.

---

> > ### Author Rebuttal · Reviewer_E5tK · 2026-04-01
> >
> > I appreciate the clarification, and I am happy to raise my scores.

---

> > > ### Author Response · Authors · 2026-04-02
> > >
> > > We sincerely thank you for the thoughtful review and for confirming that our responses have addressed all concerns. We will incorporate the additional details discussed  into the camera-ready version.

---

### Decision · Program_Chairs · 2026-04-30

**Decision:**

Accept (regular)

**Comment:**

This paper introduces ImpText-Bench, a benchmark for evaluating MLLMs on implicit text recognition in adversarial content safety scenarios, along with ImpText-Reader, a tool-augmented training framework that substantially outperforms existing baselines. Two reviewers recommended acceptance, recognizing the practical significance of the benchmark and the soundness of the evaluation, while one reviewer raised concerns about task novelty (arguing ITR reduces to OCR robustness), dataset scale, and limited methodological contribution; the authors addressed most concerns through additional experiments and clarifications, though the dissenting reviewer maintained their position. Given that the core contributions, namely revealing real-world vulnerabilities in current MLLMs and providing a carefully curated benchmark, are well-supported and the remaining objection is largely a matter of task framing rather than technical substance, I recommend acceptance, with the suggestion that the authors better clarify the distinction between ITR and OCR robustness in the final version and include the threshold sensitivity analysis in the main paper.